# Combination Therapy with Aminoglycoside in Bacteremiasdue to ESBL-Producing Enterobacteriaceae in ICU

**DOI:** 10.3390/antibiotics9110777

**Published:** 2020-11-04

**Authors:** Lucie Benetazzo, Pierre-Yves Delannoy, Marion Houard, Frederic Wallet, Fabien Lambiotte, Anne Vachée, Christian Batt, Nicolas Van Grunderbeeck, Saad Nseir, Olivier Robineau, Agnès Meybeck

**Affiliations:** 1Service de Réanimation et Maladies Infectieuses, Centre Hospitalier de Tourcoing, 135 Rue du Président Coty, 59200 Tourcoing, France; lbenetazzo@ch-lens.fr (L.B.); pydelannoy@ch-tourcoing.fr (P.-Y.D.); 2Service de RéanimationMédicale, CHRU de Lille, 2 Avenue Oscar Lambret, 59000 Lille, France; marion.houard@chru-lille.fr (M.H.); saad.nseir@chru-lille.fr (S.N.); 3Laboratoire de Microbiologie, CHRU de Lille, 2 Avenue Oscar Lambret, 59000 Lille, France; frederic.wallet@chru-lille.fr; 4Service de Réanimation, Centre Hospitalier de Valenciennes, 114 Avenue Desandrouin, 59300 Valenciennes, France; lambiotte-f@ch-valenciennes.fr; 5Laboratoire de Microbiologie, Centre Hospitalier de Roubaix, 11 Boulevard Lacordaire, 59100 Roubaix, France; anne.vachee@ch-roubaix.fr; 6Service de Réanimation, Centre Hospitalier de Dunkerque, Avenue Louis Herbeaux, 59240 Dunkerque, France; cbatt@ch-dunkerque.fr; 7Service de Réanimation, Centre Hospitalier de Lens, 99 Route de la Bassée, 62300 Lens, France; nvangrunderbeeck@ch-lens.fr; 8Service des Maladies Infectieuses et du Voyageur, Centre Hospitalier de Tourcoing, 135 Rue du Président Coty, 59200 Tourcoing, France; orobineau@ch-tourcoing.fr

**Keywords:** antimicrobialcombination, efficacy of combinations, aminoglycoside, bloodstream infections, extended-spectrum β-lactamase producing Enterobacteriaceae, critical care, bacteremia

## Abstract

Objectives: Evaluation of the efficacy of empirical aminoglycoside in critically ill patients with bloodstream infections caused by extended-spectrum β-lactamase producing Enterobacteriaceae (ESBL-E BSI). Methods: Patients treated between 2011 and 2018 for ESBL-E BSI in the ICU of six French hospitals were included in a retrospective observational cohort study. The primary endpoint was mortality on day 30. Results: Among 307 patients, 169 (55%) were treated with empirical aminoglycoside. Death rate was 40% (43% with vs. 39% without aminoglycoside, *p* = 0.55). Factors independently associated with death were age ≥70 years (OR: 2.67; 95% CI: 1.09–6.54, *p* = 0.03), history of transplantation (OR 5.2; 95% CI: 1.4–19.35, *p* = 0.01), hospital acquired infection (OR 8.67; 95% CI: 1.74–43.08, *p* = 0.008), vasoactive drugs >48 h after BSI onset (OR 3.61; 95% CI: 1.62–8.02, *p* = 0.001), occurrence of acute respiratory distress syndrome (OR 2.42; 95% CI: 1.14–5.16, *p* = 0.02), or acute renal failure (OR 2.49; 95% CI: 1.14–5.47, *p* = 0.02). Antibiotherapy appropriateness was more frequent in the aminoglycoside group (91.7% vs. 77%, *p* = 0.001). Rate of renal impairment was similar in both groups (21% vs. 24%, *p* = 0.59). Conclusions: In intensive care unit (ICU) patients with ESBL-E BSI, empirical treatment with aminoglycoside was frequent. It demonstrated no impact on mortality, despite increasing treatment appropriateness.

## 1. Introduction

Bloodstream infections (BSI) caused by extended-spectrum β-lactamase producing Enterobacteriaceae (ESBL-E) are associated with high rates of treatment failure and mortality, especially when appropriate antimicrobial therapy is delayed [1,2]. Selection of empirical antibiotic treatment is determinant in the case of severe sepsis or septic shock. Standard of care of ESBL-E infections is carbapenem. However, increase of carbapenem-resistant Enterobacteriaceae (CRE) resulting from wide empirical use of carbapenems has focused attention on the promotion of carbapenem sparing strategies [3]. Therapeutic options other than carbapenems such as third generation cephalosporins (3GC) or betalactam/betalactamase inhibitor combinations (BLBLI) could be prescribed in the case of susceptible ESBL-producing Enterobacteriaceae. However, clinical data concerning the efficacy of alternative antibiotics for the treatment of infection due to ESBL-producing Enterobacteriaceae are discordant [4,5,6]. Combination therapy with a beta-lactam and an aminoglycoside has been proposed as an alternative therapy in clinical situations at high risk of complications such as febrile neutropenia. Aminoglycosides may retain activity even in multidrug-resistant Gram-negative bacteria. Their empirical prescription will broaden the antimicrobial spectrum and reduce the risk of inappropriate therapy. Moreover, synergism between aminoglycosides and specific beta-lactams has been shown in vitro for Gram-negative bacteria. Synergistic bactericidal activity may be of particular interest in the case of severe sepsis or septic shock. Nevertheless, several recent studies have failed to demonstrate a clinical benefit of a beta-lactam and aminoglycoside combination over a single beta-lactam antibiotic [7]. Furthermore, combination therapy was associated with an increased risk of renal failure. However, few data are reported in the literature specifically concerning sepsis or septic shock due to ESBL-E infection, a clinical situation in which prescribing an aminoglycoside could benefit the prognosis. In lightof conflicting data regarding the beneficial effect of combination therapy with aminoglycoside, we conducted a retrospective study to describe antibiotic prescriptions and evaluate the prognostic impact of the initial prescription of aminoglycoside in patients with BSI caused by ESBL-E in ICU.

## 2. Patients and Methods

### 2.1. Setting and Study Population

We conducted a multicenter retrospective cohort study in the ICUs of six hospitals (Dunkerque, Lens, Lille, Roubaix, Tourcoing, and Valenciennes) in the north of France over a period of seven years (1 January 2011 through 1 January 2018). Lille Hospital is an academic hospital. All other hospitals are general hospitals. All consecutive patients treated for ESBL-E BSI during the study period were retrospectively included.

Cases were identified using a laboratory database query completed with ICU clinical databased analysis. Cases were defined as adults with blood culture(s) yielding ESBL-E within 24 h prior to ICU admission or during the ICU stay. In the case of several positive ESBL-E bacteremias during the same infectious episode, only the first positive blood culture was considered for analysis. Patients who had multiple episodes of ESBL-E BSI were included only once in the analysis, and subsequent episodes were considered recurrences or re-infections.

Enterobacteriaceae identification and in vitro antimicrobial susceptibility testing were performed with a Vitek 2 system (bioMérieux, Marcy l’Etoile, France). ESBL diagnosis and susceptibility testing were carried out according to the European Committee on Antimicrobial Susceptibility Testing (EUCAST) breakpoints [8].

Our study was carried out in accordance with national guidelines concerning an observational study conducted retrospectively on collected data (article R.1121-1-1, Décret no. 2017-884 du 9 mai 2017). The present study obtained ethical approval from the local ethicscommittee ofDron Hospital (comitéd’éthique du Centre Hospitalier Gustave Dron) Ethic code: CNIL 2019-10.

### 2.2. Data Collection and Definitions

The following data were recorded: demographic characteristics, indication(s) of ICU admission, underlying clinical conditions, immunodeficiency, and severity of illness at admission. The underlying diseases were classified with criteria proposed by McCabe and Jackson [9]. Immunodeficiency was defined as neoplasia, neutropenia (neutrophil count <0.5 × 10^9^ cells/L), treatment with glucocorticosteroids, and/or other immunosuppressive therapy, AIDS, as defined by the U.S. Centers for Disease Control and Prevention. Severity of illness was assessed by Simplified Acute Physiology Score II (SAPS II) and Sepsis-related Organ Failure Assessment (SOFA) score [10,11]. At the time of BSI diagnosis, we recorded prior antimicrobial therapy within one month before BSI, prior ESBL-E colonization, duration of hospital and ICU stay before BSI, severity of illness, presence of shock, usual biochemical, and hematological tests. Shock was defined by usual criteria [12].

Antimicrobial prescriptions were recorded. Empirical treatment was defined as the prescription of antibiotics before culture results were available; empirical treatment was considered appropriate when the isolated pathogen was susceptible in vitro to at least one of the empirically administered antibiotics according to the EUCAST breakpoints. Timing of empirical treatment was considered adequate when it was started during the first 24 h after BSI diagnosis. Definitive treatment was defined as start, continuation, or change to an effective antibiotic treatment after the culture result was available, according to the pathogen’s susceptibility pattern. De-escalation included switching from combination to monotherapy, or from one beta-lactam to another one with a narrower spectrum and lighter selective pressure according to a six-rank consensual classification of beta-lactams [13]. Combination therapy was only considered if two or three antibiotics active against *Enterobacteriaceae* were part of the initial empirical treatment. BLBLI combination, 3GC, and carbapenems were administered by continuous infusion after a loading dose or by extended infusions, depending on the stability of the antibiotic drugs after reconstitution. 

All patients were follow-up until death or release of ICU. Clinical cure was defined as resolution of clinical signs and symptoms, negative blood cultures, and no requirement for additional antibacterial treatment at the end of treatment. We also assessed the proportion of adequate initial antibiotic treatment, duration of mechanical ventilation, catecholamine infusion, and ICU stay after BSI onset, microbiologic success, and multidrug-resistant bacteria acquisition. Microbiologic success was defined as the eradication of the microorganism in blood cultures at the end of treatment. Documented persistence or recurrence was assessed at the end of follow-up. 

### 2.3. Statistical Analysis

Continuous variables were expressed as mean values ± standard deviation or as median (interquartile range), depending on the normality of their distribution. They were compared using the Student’s test or the Mann–Whitney U test, as appropriate. Categorical variables were expressed as percentages and evaluated using the chi-square test and Fisher’s test when appropriate. Differences between groups were considered to be significant for variables yielding a *p* value ≤ 0.05. To determine the independent effect of the variables on mortality at day 30, we calculated the corresponding unadjusted and multivariate-adjusted hazard ratios of death using the Cox proportional hazard regression analysis. All covariates with *p* < 0.2 in the unadjusted model and use of aminoglycoside and combination therapy were entered into the multivariate model. All statistical analysis were performed using SAS 9.2.

## 3. Results

### 3.1. Demographic and Clinical Data

In Figure 1, the flow chart of the study inclusion process is reported (Figure 1). Of the 334 critically ill patients selected for having ESBL-E BSI during the study period, 27 were excluded from the analysis. In four cases, data were missing with regard to empirical antimicrobial prescriptions. In 23 cases, patients received aminoglycosides only as definitive treatment. Among the 307 patients included, 169 were treated with aminoglycoside as the empirical treatment and constituted the group with aminoglycoside; 138 did not received aminoglycoside at any time and constituted the group without aminoglycoside. 

Demographic and clinical characteristics of our patients, depending on aminoglycoside prescription, are shown in Table 1. Our patients were predominantly male (67%) with a median age of 63 years (IQR, 55–70). The majority of our patients entered ICU for medical admission (83%). The median SAPS II value on ICU admission was 50 (IQR, 38–51). Patients treated with aminoglycoside were more frequently receiving immunosuppressive treatment.

ESBL-E BSI was community acquired in 34% cases. During the two months prior to the occurrence of BSI, 302 patients (98%) had received antibiotics. One hundred and seventy two (56%) patients were colonized with ESBL-E. Upon BSI onset, 164 (53%) patients exhibited shock: 96 (57%) in the aminoglycoside group and 59 (43%) in the non-aminoglycoside group (*p* =0.02). Source of infection was mainly pulmonary in 43% of cases. Joint and bone infections were few, but more frequent in the group without aminoglycoside (3.6% vs. 0%, *p* =0.02).

### 3.2. Microbiological Data

Overall, *Klebsiella pneumoniae* (60%), *Enterobacter* sp. (20%), and *Escherichia coli* (16%) were the most frequently involved pathogens. Comparison between groups (with aminoglycoside vs. without aminoglycoside) found no difference. Proportion ofESBL-E sensitive to aminoglycosides was 55%.

### 3.3. Empirical and Definitive Antibiotherapy

Table 2 shows a comparison of antibiotics used in empirical and definitive regimens among patients treated with or without aminoglycoside. Among our patients, 231 (75%) received a combination therapy, with aminoglycoside in 166 patients (54%). One hundred and fifty-eight patients (51%) received a beta-lactam plus an aminoglycoside, five (2%) colistine plus an aminoglycoside, and three (1%) fluoroquinolone plus an aminoglycoside. A carbapenem was prescribed in 184 patients (60%) with the same frequency in both groups (64% with aminoglycoside vs. 57% without aminoglycoside; *p* =0.55).

The proportion of patients who received appropriate initial antibiotic therapy was 79% (91% with aminoglycoside vs. 77% without aminoglycoside, *p* = 0.001). In 26 patients, the aminoglycoside was the only active antibiotic.

Adjustment of initial empirical antibiotherapy was performed in 39% of patients treated initially without aminoglycoside and in 49% of patients treated with aminoglycoside (*p* = 0.11). Switch to an alternative to carbapenem was performed in 4% of patients in the group without aminoglycoside and 14% of patients in the group with aminoglycoside (*p* = 0.075). 

Among patients treated initially without aminoglycoside, 14% were transitioned to an aminoglycoside containing regimen.

### 3.4. Aminoglycoside Use and Impact

The most frequently prescribed aminoglycoside was amikacin in 83% of patients treated with aminoglycoside. Mean duration of aminoglycoside treatment was 1.6 day. Drug monitoring was performed in 39% of patients treated with aminoglycoside.

Outcome depending on aminoglycoside prescription is summarized in Table 3. Death rate was similar in both groups, respectively 43% and 39% with and without empirical aminoglycoside (*p* = 0.54). Eight percent of patients treated empirically with aminoglycoside, and 11% without aminoglycoside experienced a relapse (*p* = 0.44). Persistent colonization with ESBL-*Enterobacteriaceae* was observed in 28% of patients treated with empirical aminoglycoside and 30% of patients treated without empirical aminoglycoside (*p* = 0.92). Acute renal failure occurred in 21% of patients treated empirically with aminoglycoside and in 23% of patients treated empirically without aminoglycoside (*p* = 0.59). 

### 3.5. Risk Factors for Mortality

Significant factors associated with death at day 30 in univariate and multivariate analysis are reported in Table 4. The multivariate analysis identified age ≥70 years (OR 2.67, 95% CI 1.09–6.54, *p* = 0.03), history of solid organ transplantation (OR 5.2, 95% CI 1.4–19.35, *p* = 0.01), nosocomial infection (OR 8.67, 95% CI 1.74–43.08, *p* = 0.008), need for vasoactive drugs at day 2 after BSI onset (OR 3.61, 95% CI 1.62–8.02, *p* = 0.001), acute respiratory distress syndrome (ARDS) (OR 2.42, 95% CI 1.14–5.16, *p* = 0.02), or acute renal failure occurrence (OR 2.49, 95% CI 1.14–5.47, *p* = 0.02) as the variables independently associated with mortality. The prescription of empirical aminoglycoside (OR 1.05, 95% CI 0.54–5.47, *p* = 0.02) did not appear as an independent factor for mortality at day 30. 

## 4. Discussion

Our study confirmed the high mortality associated with ESBL-E BSI in critically ill patients. In our cohort, prescription of aminoglycoside was frequent, led to a higher proportion of appropriate initial treatment, and appeared safe. However, empirical use of aminoglycoside was not associated with a reduction of mortality.

In our cohort of critically ill patients suffering from ESBL-E BSI, the 30 day mortality rate was 40%. This rate was logically significantly higher than those observed in the previously published studies conducted both on patients in the medicine ward and in ICU [2,14]. However, Russo et al. analyzed the outcome of 354 patients suffering from ESBL-E BSI who developed severe sepsis or septic shock, and reported a similar mortality rate of 44% [15]. In a multicenter study conducted in ICU, ESBL-E infections were associated with a 1.8-fold increase in the overall hazard of dying in the ICU [16]. 

The high mortality associated with ESBL-E BSI in critically ill patients justified studies designed to identify prognosis factors and to improve their management. Factors associated with mortality in our cohort were age ≥70 years, history of solid organ transplantation, nosocomial infection, the need for vasoactive drugs at day 2 after BSI onset, and acute respiratory distress syndrome, or acute renal failure occurrence. Similarly, Russo et al. identified as factors associated with death in the case of severe sepsis or septic shock due to ESBL-E BSI, the severity of comorbidities and of initial clinical conditions (expressed by age, McCabe classification, Charlson Comorbidity Index, and Pitt bacteremia score) and the worsening of clinical conditions (expressed by the need of escalation of initial antibiotic therapy) [15]. In their study, the abdominal source of infection was also associated with an unfavorable outcome at 30 days. In contrast, the use of a quinolone in definitive therapy was associated with 30-day survival and found no association between a particular initial therapy and a reduction of mortality. 

Our study failed to demonstrate a favorable prognostic impact of initial aminoglycoside prescription in patients with BSI caused by ESBL-E in ICU. In our cohort, prescription of aminoglycoside was frequent. Almost half of our patients received aminoglycoside as part of the initial therapy, mostly as a combination therapy. This observation is concordant with the results of a French survey focused on the early management of BSI. Empirical combination therapy was used in 57% patients with severe sepsis or septic shock, frequently including an aminoglycoside [17]. In a large international retrospective cohort study including 4662 patients with septic shock, one third of the patients received initial combination therapy with an aminoglycoside in more than 10% of cases [18].

In this last study, the authors demonstrated a therapeutic benefit of early combination therapy comprising at least two antibiotics of different mechanisms with in vitro activity for the isolated pathogen in patients with bacterial septic shock. The arguments for prescribing a combination therapy with aminoglycoside in critically ill patients with ESBL-E BSI are synergistic bactericidal activity and antimicrobial spectrum broadening. The retrospective design of our study prevents any strong conclusion concerning the reason for the ICU doctors for prescribing aminoglycoside. We can only assume the rationale for their prescription by analyzing patients treated with or without aminoglycoside. In our cohort, prescription of initial antibiotic combination with aminoglycoside seemed mainly motivated by its synergistic bactericidal activity. Indeed, aminoglycoside prescription was more frequent in the case of septic shock and was not associated with carbapenem sparing, since the proportion of our patients receiving carbapenem as the initial therapy was similar with or without empirical aminoglycoside. In our study, neither prescription of aminoglycoside nor prescription of an initial active bitherapy were associated with reduced mortality. Numerous studies have compared aminoglycoside and beta-lactam combination with betalactam monotherapy [7,19,20]. Most studies did not find any survival benefit of combination therapy. The heterogeneity of population included in these studies prevented any strong conclusion [21]. Few studies focused on critically ill patients. In this particular context, combination therapy might improve the clinical prognosis of bacteremia. Delannoy PY et al. found a survival benefit of aminoglycoside combination therapy over betalactam monotherapy in the case of ICU-acquired BSI [22]. Similarly, Kumar et al. demonstrated a clinical benefit of early combination therapy in the case of septic shock [18]. None of these studies included only resistant bacteria. In our cohort, prescription of early combination therapy with an aminoglycoside resulted in aminoglycoside effective monotherapy in 16% of cases, preventing the synergistic bactericidal activity in those cases. In a multinational retrospective cohort study of patients with ESBL-E BSI, Palacios-Baena et al. did not find different outcomes in patients receiving empiric active monotherapy with aminoglycoside compared with carbapenem [23].

In our cohort, antimicrobial spectrum broadening induced by aminoglycoside prescription led to a significantly higher proportion of appropriate initial antibiotherapy in patients receiving aminoglycoside. Appropriate therapy for sepsis has been shown to improve survival, especially in critically ill patients [24,25,26]. In a study including patients with ESBL-E BSI, Tumbarrelo et al. identified the absence of adequate antimicrobial therapy within the first 72 h of infection as an independent predictor of mortality [2]. The INCREMENT-ESBL predictive score validated to evaluate the risk of death in patients with ESBL-E BSI includes appropriate early targeted therapy [27]. In our cohort of critically ill patients suffering from ESBL-BSI, appropriate initial antibiotherapy was not associated with mortality reduction. Similarly, Russo et al. found no impact on mortality of adequate initial therapy in their series of severe sepsis and septic shock associated with ESBL-E BSI [15]. They advocated that, in the case of septic shock, appropriate antimicrobial treatment failed to stop the lethal cascade of events already triggered. Rapid diagnosis of sepsis and the quality of supportive care through implementation of sepsis guidelines are key to prognosis [28].

The potential clinical benefit of aminoglycoside combination therapy is counterbalanced by aminoglycoside side effects, especially nephrotoxicity. Aminoglycoside nephrotoxicity has been shown to be influenced by multiple factors including longer duration of therapy, preexisting renal or liver disease, shock, older age, and location in intensive care [29]. In our cohort of critically ill patients with ESBL-E BSI, we found no deleterious impact of aminoglycoside prescription on acute renal failure occurrence and need for hemodialysis. Duration of aminoglycoside use was short, with a mean duration of treatment of 1.6 day. Almost two-thirds of our patients received only a single dose of aminoglycoside. Guidelines to use a short course of once daily high dosing of aminoglycoside are quite recent [30], but are widely applied by French physicians [31]. Few studies have analyzed nephrotoxicity of empirical antibiotic therapy including a short-course of aminoglycoside. Two retrospective studies found no association between exposure to aminoglycosides and acute renal failure [32,33]. One prospective study conducted in an ICU found an increased incidence of acute renal failure in patients treated with a short-course of aminoglycoside therapy as part of empirical combination therapy in patients with severe sepsis and septic shock [34]. Discordant results concerning nephrotoxicity and inconsistent prognosis impact of initial aminoglycoside therapy in critically ill patients are in contrast with their broad prescription, calling for more robust studies including randomized clinical trials. Critically ill patients with sepsis, potentially caused by Gram-negative bacteria could be included. Randomization should be stratified according to the risk of infection with ESBL-E.

Our study has several limitations. Asour sample was relatively small, a type II error is possible and studies with larger study population might demonstrate additional differences.

We conducted a retrospective multicentric observational study. Due to the retrospective nature of our study, patients receiving combination aminoglycoside therapy may differ in some systematic way beyond the direct effects of the treatment strategy. However, multivariateanalysis allowed for adjusting for confounding variables. Finally, underdosing of aminoglycosides may also have contributed to the absence of their clinical benefit.In our cohort, therapeutic drug monitoring of aminoglycosides was performed in less than half of the patients. Several previous studies conducted in critically ill patients have shown that evena high dose regimen of aminoglycosides led to pharmacodynamic target achievement in less than two thirds of the patients [35,36]. Early achievement of optimal aminoglycoside concentration was associated with better clinical and microbiological responses [36]. Future prospective studies should use high doses of amioglycosides and drug monitoring to confirm achievement of adequate peak concentrations.

## 5. Conclusions

In our cohort of critically ill patients with ESBL-E BSI, prescription of initial combination therapy with aminoglycoside was frequent. However, its benefit was not clearly demonstrated. Aminoglycoside prescription increased the proportion of appropriate empirical therapy, but was not associated with a reduction inmortality. Prospective studies are needed to better determine prognosis impact of initial aminoglycoside therapy in critically ill patients with suspected ESBL-E BSI. They should target a high-risk population as identified in our study, particularly immunodepressed patients and patients suffering from hospital acquired infection.

## Figures and Tables

**Figure 1 antibiotics-09-00777-f001:**
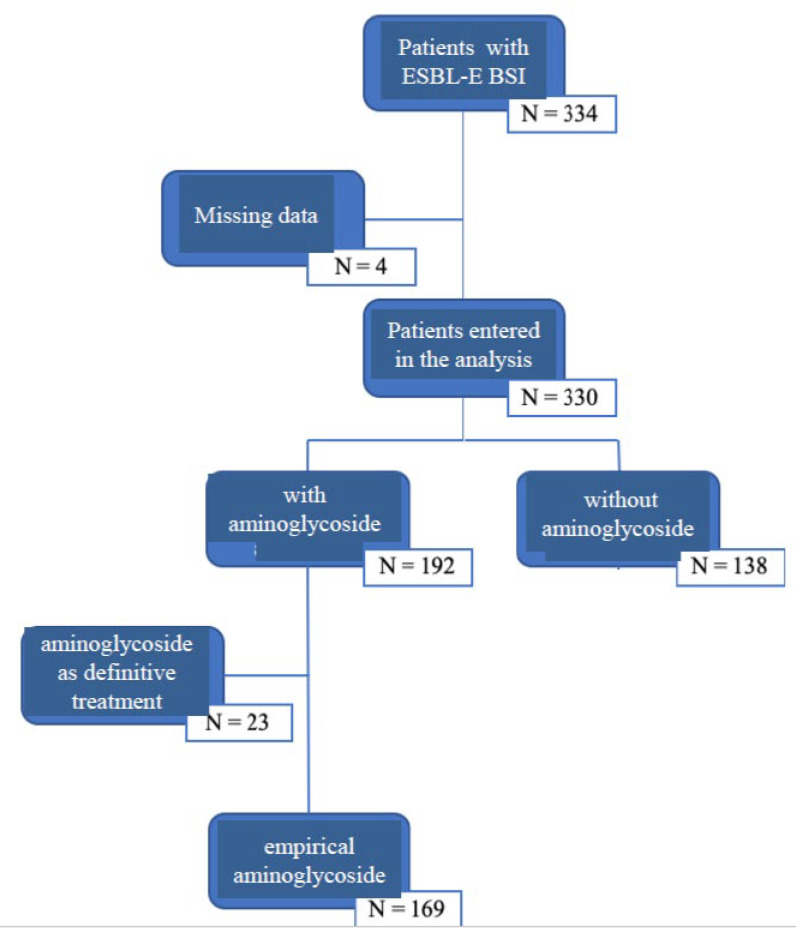
Flow chart of the study inclusion process.

**Table 1 antibiotics-09-00777-t001:** Demographic and clinical characteristics of our patients depending on aminoglycoside prescription.

Characteristics	Without Aminoglycoside *n* = 138	With Aminoglycoside *n* = 169	*p*
Male sex, *n* (%)	95 (68.8)	109 (64.5)	0.496
Age (years)	63 (54–70)	62 (55–70)	0.775
McCabe^3^ 2	98 (71)	117 (69)	0.840
SOFA score	7 (4.25–10.0)	8 (5–11)	0.154
Comorbidity			
- Diabetes	42 (30.4)	49 (29.2)	0.908
- Renal insufficiency	16 (11.6)	18 (10.7)	0.937
Immunodeficiency	49 (35.5)	83 (49.1)	0.020
- Immunosuppressive therapy in the last 3 months	7 (5.1)	25 (14.8)	0.010
- Transplantation	6 (4.3)	16 (9.5)	0.132
- Solid cancer	22 (15.9)	22 (13)	0.573
- Hematological malignancy	14 (10.1)	20 (11.8)	0.775
Admission			
- Medical	110 (79.7)	145 (85.8)	0.207
- Scheduled surgical	2 (1.4)	0 (0)	0.201
- Unscheduled surgical	14 (10.1)	20 (11.8)	0.775
Community acquired infection	42 (30.4)	51 (30.2)	0.999
Origin of the infection			
- Urinary tract	13 (9.4)	25 (14.8)	0.202
- Intra-abdominal infection	24 (17.4)	29 (17.2)	0.999
- Catheter related infection	32 (23.2)	30 (17.8)	0.315
- Respiratory tract	53 (38.4)	75 (44,4)	0.313
- Bone infection	5 (3.6)	0 (0)	0.018
- other	10	6	0.308
Etiology			
- *Klebsiella* sp.	87 (63)	97 (57.4)	
- *Enterobacter* sp.	30 (21.7)	31 (18.3)	
- *E. coli*	19 (13.8)	33 (19.5)	
- Other ^§^	2 (1.5)	8	
polymicrobial infection	25 (1.1)	30 (17.8)	0.999

All numbers represent the number of patients (percent of total in treatment arm) except otherwise specified. ^§^
*Serratia* sp. (*n* = 5), *Proteus* sp. (*n* = 3), *Citrobacter* sp. (*n* = 2).

**Table 2 antibiotics-09-00777-t002:** Comparison of antibiotics used in the empirical regimen among patients treated with or without aminoglycoside.

Antibiotics	Without Aminoglycoside *n* = 138	With Aminoglycoside *n* = 169	*p*
	*n* (%)	*n* (%)	
Non carbapenem-betalactams	50 (36)	58 (34)	0.770
- amoxicillin-clavulanate	2 (1.4)	0 (0)	
- ticarcillin-clavulanate	1 (0.7)	0 (0)	
- piperacillin-tazobactam	29 (21)	39 (23)	
- cefotaxime/ceftriaxone	5 (3.6)	6 (3.6)	
- cefepime	4 (2.9)	3 (1.8)	
- ceftazidime	5 (3.6)	8 (4.8)	
- ceftazidime-avibactam	1 (0.7)	0 (0)	
- ceftolozane-tazobactam	1 (0.7)	0 (0)	
- other	2 (1.4)	1 (0.6)	
Carbapenem antibiotics	78 (56.5)	106 (64)	0.546
- imipenem	58 (42)	79 (47)	
- meropenem	16 (11.6)	18 (11)	
- ertapenem	4 (2.9)	8 (4.7)	
- doripenem	0 (0)	1 (0.6)	
Fluoroquinolone	35 (25.3)	6 (3.6)	<0.001
- ofloxacin/levofloxacin	6 (4.3)	2 (1.2)	
- ciprofloxacin	29 (21)	4 (2.4)	
Anti-cocci Gram positive	4 (2.9)	8 (4.7)	0.560
- vancomycin	2 (1.4)	6 (3.6)	
- teicoplanin	1 (0.7)	2 (1.2)	
- daptomycin	1 (0.7)	0 (0)	
Other	36 (26)	11 (6.5)	<0.001
- monobactam	1 (0.7)	1 (0.6)	
- trimethoprim-sulfamethoxazole	2 (1.4)	0 (0)	
- metronidazole	2 (1.4)	2 (1.2)	
- colistin	31 (22.5)	8 (4.7)	

**Table 3 antibiotics-09-00777-t003:** Outcome according to empirical aminoglycoside prescription.

Outcome	Without Aminoglycoside *n* = 138	With Aminoglycoside *n* = 169	*p*
	*n* (%)	*n* (%)	
Complications:			
Septic shock	59 (42.8)	96 (56.8)	0.020
Acute respiratory distress syndrome (ARDS)	27 (19.6)	33 (19.5)	1.000
Acute renal failure	33 (23.9)	35 (20.7)	0.593
Disseminated intravascular coagulation (DIC)	15 (10.9)	11 (6.5)	0.246
Bacteremia relapse	15 (11.1)	13 (7.9)	0.448
Colonization with multi-drug resistant bacteria	39 (29.5)	47 (28.3)	0.917
Colonization with carbapenem-resistant enterobacteriaceae (CRE)	4 (2.9)	2 (1.2)	0.414
fungemia	3 (2.1)	3 (1.8)	1.000
*Clostridium difficile* colitis	2 (1.4)	2 (1.2)	1.000
Evolution:			
SOFA day 1 > 7	62 (44.9)	89 (53)	0.487
Vasopressors > 48 h	31 (23)	46 (28)	0.24
Mechanical ventilation at day 30	22 (16)	22 (13)	0.55
Mortality at day 30	59 (42.8)	66 (39.3)	0.545
Death in ICU	63 (45.7)	73 (43.2)	0.752
- Male sex	48%	42%	0.206
- Age ≥ 70 years	58%	46%	0.291
- Immunodepression	54%	58%	0.820
- Hospital acquired	54%	47%	0.336
- Non urinary tract infection	51%	44%	0.274
- SOFA ≥ 5	48%	45%	0.699
- McCabe ≥ 2	60%	54%	0.402

**Table 4 antibiotics-09-00777-t004:** Bivariate and multivariate analysis of risk factors associated with 30-day mortality.

Variables	Bivariate Analysis		Multivariate Analysis	
	OR	*p*	OR	*p*
Aminoglycoside	0.84 [0.53–1.34]	0.47	1.05 [0.54–2.06]	0.89
Male sex	0.91 [0.56–1.46]	0.69	0.50 [0.25–1.01]	0.05
Age (years) (reference:age <55)				
55 ≤ Age < 62	1.75 [0.90–3.39]	0.10	1.64 [0.67–4.03]	0.28
62 ≤ Age < 70	1.55 [0.80–2.97]	0.19	1.49 [0.60–3.65]	0.39
Age ≥ 70	2.78 [1.44–5.37]	0.00	2.67 [1.09–6.54]	0.03
Medical admission	0.62 [0.33–1.16]	0.13	0.72 [0.28–1.88]	0.51
Respiratory insufficiency	0.97 [0.57–1.63]	0.90		
Chronic liver insufficiency	0.94 [0.41–2.16]	0.88		
Cardiac insufficiency	2.16 [1.07–4.37]	0.03	2.16 [0.78–5.98]	0.14
Diabetes	1.04 [0.64–1.68]	0.89		
Solid cancer	1.27 [0.67–2.42]	0.46		
Transplantation	3.79 [1.52–9.40]	0.004	5.20 [1.4–19.35]	0.01
Source (reference:urinary tract)				
Respiratory tract	1.93 [0.88–4.23]	0.10		
Catheter related infection	0.91 [0.37–2.20]	0.83		
Intra-abdominal infection	3.1 [1.27–7.60]	0.01		
Other	2.88 [0.93–8.88]	0.07		
Hospital acquired infection	3.82 [1.28–11.44]	0.02	8.67 [1.74–43.08]	0.01
SOFA (reference: SOFA < 5)				
5 ≤ SOFA < 7	0.92 [0.47–1.79]	0.80	0.54 [0.21–1.42]	0.21
7 ≤ SOFA < 11	1.15 [0.65–2.01]	0.63	0.52 [0.23–1.18]	0.12
SOFA ≥ 11	2.32 [1.20–4.48]	0.01	1.69 [0.66–4.34]	0.28
Duration of vasopressors (reference: <24 h)				
between 24 and 48 h	4.21 [2.33–7.58]	<0.001	3.02 [1.24–7.31]	0.01
>48 h	4.09 [2.30–7.27]	<0.001	3.61 [1.62–8.02]	0.002
Polymicrobial infection	1.25 [0.70–2.24]	0.45		
Active combination therapy	0.82 [0.50–1.37]	0.45	0.55 [0.28–1.08]	0.08
Initial appropriate antibiotherapy	1.16 [0.59–2.29]	0.67		
ARDS	3.23 [1.81–5.76]	<0.001	2.42 [1.14–5.16]	0.02
Acute renal failure	4.87 [2.72–8.72]	<0.001	2.49 [1.14–5.47]	0.02

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
