# Peer review of "Combination Therapy with Aminoglycoside in Bacteremiasdue to ESBL-Producing Enterobacteriaceae in ICU"

_antibiotics, 2020, doi:10.3390/antibiotics9110777_

Round 1
Reviewer 1 Report
I read this article by L Benetazzo et al. with great interest. It provides unique insight into the outcomes of patients with ESBL-producing Enterobacteriaceae bloodstream infections (ESBL-E BSI) in a large cohort of ICU patients. Further, it provides some evidence that perhaps combination therapy with aminoglycosides are not beneficial in proven ESBL-E BSI infections with some caveats.
Major Comments:
- It would be helpful to see an outcomes (mortality) analysis divided by initial clinical characteristics shown in Figure 1, to understand if there are potential subgroups who may benefit from combination therapy. This could be included in Table 3 or as a separate table.
Minor Comments:
- In Figure 1, the labels should be made larger as they are difficult to read.
- In Figure 1, on the flowchart to the right, it does not seem necessary to connect the “Without Aminoglycoside” box to the subsequent “Without Aminoglycoside” box as there are no alternative paths. If any patients were transitioned to an aminoglycoside in that treatment group, this should be included here.
- In Line 182, the authors write that the majority of patients were admitted to the ICU for “Medical Diagnosis.” The alternative to “medical diagnosis” is unclear. Additionally, the authors change this to “Medical admission” in Table 1. These labels should be consistent and alternatives should be included.
- The authors note in the methods section that “immunodeficiency” is defined by neoplasia, neutropenia, glucocorticoids/other immunosuppressive therapies, and AIDS. This is inconsistent with Table 1 as Neoplasia is included in “comorbidity.” Please clarify this disconnect in the text and/or the Table. I assume that neoplasia (both hematological and solid organ) are included both as a comorbidity and immune-suppression based upon the methods section.
- There seems to be a propensity for clinicians to choose dual therapy for these patients. Please include combined immune-suppression (Neoplasms, Immunosuppressive therapies, transplantation, neutropenia) in Table 1 to address this question.
- In table 1, please clarity the meaning of Immunosuppressive Therapy < 3 months. Are patients who were immune-suppressed for > 3 months included?
- In Table 1, Plurimicrobial should read Polymicrobial.
- In Table 1, please include the meaning of the numbers in parentheses. I presume this is % (except for SOFA score), but this should be explicitly noted in the table legend.
- In Table 1, please include a description of the “other” category under “etiology” with an asterisk.
- In the results section, Lines 226-233, it would be helpful to know how many patients of the 23% who did not receive an aminoglycoside initially were transitioned to an aminoglycoside, even if none were transitioned to aminoglycosides.
- There are multiple spelling errors in Table 4 that should be resolved.
- In Line 355, would change “extra-renal epuration” to “hemodialysis,” assuming this is the meaning of the statement.
- In their conclusions section, I recommend that the authors include a statement about consideration of targeting high risk patients as identified in their study (identified by their multivariate analyses in Table 4) with empirical aminoglycoside therapy for future prospective trials.
Author Response
Reviewer 1.
Major Comments :
- It would be helpful to see an outcomes (mortality) analysis divided by initial clinical characteristics shown in Figure 1, to understand if there are potential subgroups who may benefit from combination therapy. This could be included in Table 3 or as a separate table.
Minor Comments :
- In Figure 1, the labels should be made larger as they are difficult to read.
The authors modified Figure 1 as suggested by the reviewer.
- In Figure 1, on the flowchart to the right, it does not seem necessary to connect the « Without Aminoglycoside » box to the subsequent « Without Aminoglycoside » box as there are no alternative paths. If any patients were transitioned to an aminoglycoside in that treatment group, this should be included here.
The authors modified Figure 1 as suggested by the reviewer.
- In Line 182, the authors write that the majority of patients were admitted to the ICU for « Medical diagnosis ». The alternative to « medical diagnosis » is unclear.
Additionally, the authors change this to « Medical admission » in Table 1. These labels should be consistent and alternatives should be included.
The labels referred to type of admission included in SAPS II score : scheduled surgical, medical, unscheduled surgical. The authors changed the text in : « The majority of our patients entered ICU for medical admission (83%).»
The authors added in Table 1 the proportion of scheduled surgical and unscheduled surgical admissions.
- The authors note in the method section that « immunodeficiency » is defined by neoplasia, neutropenia, glucocorticoids/other immonusuppressive therapies, and AIDS. This is inconsistent with Table 1 as Neoplasia is included in « comorbidity ». Please clarify this disconnect in the text and/or the Table. I assume that neoplasia (both hematological and solid organ) are included both as comorbidity and immunesuppression based upon the methods section.
The authors modified Table 1 as suggested by the reviewer. Neoplasia (both hematological and solid organ) were included as « immunodeficiency ».
- There seems to be a propensity for clinicians to choose dual therapy for these patients. Please include combine immune-suppression (Neoplasms, Immunosuppressive therapies, transplantation, neutropenia) in Table 1 to address this question.
The authors included combine immune-suppression in Table 1.
- In table 1, please clarity the meaning of Immunosuppressive Therapy < 3 months. Are patients who were immune-suppressed for > 3 months included?
The authors clarified the meaning of immunosuppressive therapy <3 months. They replaced the terms « immunosuppressive therapy <3 months » by « immunosuppressive therapy in the last 3 months ».
- In table 1, Plurimicrobial should read Polymicrobial.
The authors changed Plurimicrobial for Polymicrobial.
- In Table 1, please include the meaning of the numbers in parentheses. I presume this is % (except for SOFA score), but this should be explicitly noted in the table legend.
The authors precised the meaning of the numbers in parentheses in the table legend :
All numbers represent number of patients (percent of total in treatment arm) except where otherwise specified.
- In Table 1, please include a description of the « other » category under « etiology » with an asterisk.
The authors included a description of the « other » category under « etiology » with an asterisk :Serratia sp. (n=5), Proteus sp. (n=3), Citrobacter sp. (n=2).
- In the results section, Lines 226-233, it would be helpful to know how many patients of the 23% who did not receive an aminoglycoside initially were transitioned to an aminoglycoside, even if none were transitioned to aminoglycosides :
A total of 161 patients did not received initial treatment with aminoglycoside. Among them, 23 (14%) were transitioned to an aminoglycoside containing antibiotic regimen as definitive treatment. The authors precised this point in the results section : « Among patients treated initially without aminoglycoside, 14% were transitioned to an aminoglycoside containing regimen ».
- There are multiple spelling errors in Table 4 that should be resolved.
The authors corrected the spelling errors in Table 4.
- In Line 355, would change « extra-renal epuration » to « hemodialysis » assuming this is the meaning of the statement.
The authors changed « extra-renal epuration » to « hemodialysis ».
- In their conclusion section, I recommend that the authors include a statement about consideration of targeting high risk patients as identified in their study (identified by their multivariate analyses in Table 4) with empirical aminoglycoside therapy for future prospective trial.
The authors included in their conclusion a statement about consideration of targeting high risk patients as identified in their study :
« They should target high risk population as identified in our study, particularly immunodepressed patients, and patients suffering from hospital acquired infection.
Reviewer 2 Report
This is a retrospective study of the usefulness of aminoglycoside in patients with ESBL-E BSI in the ICU, which is interesting, however, I have some comments to the author.
1. Regarding the “empirical use of aminoglycoside”, I thought it would be better if I could understand the reason and/or rationale for the ICU doctors in charge of using aminoglycoside.
2. I could not understand the meaning of the number "Source" in Table 1, 13 (9.4) & 25 (14.8). What do they mean?
3. Similarly, in Table 1 “bone infection”, without aminoglycoside is 3.6%, but in the description in the text, it is 3.7%, which is inconsistent.
4. P -value is not listed in "Anti-cocci Gram positive" & "Other" in Table 2.
Author Response
Reviewer 2.
- Regarding the « empirical use of aminoglycoside », I thought it would be better if I could understand the reason and/or rationale for the ICU doctors in charge of using aminoglycoside.
The authors precised the limits of their study : « The retrospective design of our study prevents any strong conclusion concerning the reason for the ICU doctors for prescribing aminoglycoside. We can only assume the rationale for their prescription by analysing patients treated with or without aminoglycoside. »
- I could not understand the meaning of the number « Source »in Table 1, 13 (9.4) and 25 (14.8). What do they mean ?
The rows in Table 1 were shifted during the layout. The authors made the proper corrections.
- Similarly, in Table 1 « bone infection », without aminoglycoside is 3.6%, but in the description in the text, it is 3.7%, which is inconsistent.
The authors corrected the result in the text : « 3.6% ».
- P-value is not listed in « Anti-cocci Gram positive » and « other » in Table 2.
The authors listen P-value in « Anti-cocci Gram positive » and « other » in Table 2.
Reviewer 3 Report
The manuscript titled- "Combination therapy with aminoglycoside in bacteraemias due to ESBL-producing Enterobacteriaceae in ICU" by Benetazzo L et. al. presents a retrospective study with 307 ICU patients afflicted with ESBL-E BSI and conclude that prescription of initial combination therapy was not necessarily beneficial even though it was widely prescribed. They also conclude that this approach didn't present any significant reduction in mortality. The authors also suggest that prospective studies may offer a clearer picture in determining the impact of initial aminoglycoside therapy in suspected ESBL-E BSI, esp. that are critically ill. Overall, I find that the manuscript is relevant and packaged well with lucid data presentation in tabular form, the only caveat I can find is not having sufficiently large n but the authors do acknowledge this limitation. There is no doubt that the study is clinically relevant and the conclusions made from the data are sound.
Thus, I recommend this article in Antibiotics.
Author Response
The authors have no comment.
Reviewer 4 Report
The manuscript ID: antibiotics-963589 by Benetazzo and colleagues reports a retrospective study on the efficacy of the empirical prescription of combination therapies, including aminoglycosides, in the treatment of bloodstream infections affecting patients in intensive care units in six French hospitals.
After evaluating different parameters (including demographic information, clinical data, antibiotic treatment and its outcome and important risk factors), the authors stated that the inclusion of aminoglycosides in combination therapy, although resulting appropriate, doesn’t affect the survival of patients in severe hill conditions and suggest further prospective studies on this subject.
The study is clear, the results are well presented and supported by both statistical significance and literature data, the conclusions are logical and sounding. Moreover, they highlight the need for a constant evaluation of the therapeutic protocols adopted in specific categories of patients (in this case those in intensive care units) and for the design of novel and more efficient approaches to counteract antibiotic resistant bacterial infections.
As presenting data about the clinical impact of combination therapies in the clinical practise, the paper fits in the proposed special issue and there are no concerns for its publication in “Antibiotics”.
Minor comments:
-Line 36, please correct “evaluation of the efficacy of empirical aminoglycosides”;
-Please delete line 41;
-Line 111, please correct “the present study has got ethical approval or was approved”;
-Table 4, please correct “Male sex”;
-Line 314, please correct “carbapenem spare”.
Author Response
Reviewer 4.
Minor comments :
- Line 36, please correct « evaluation of the efficacy of empirical aminoglycosides »
The authors made the correction.
- Please delete line 41 :
The authors deleted line 41
- Line 111, please correct « the present study has got ethical approval or was approved ».
- The authors made the suggested correction : « The present study has got ethical approval from the local ethical committee of Dron Hospital... ».